# The potential impacts of vector host species fidelity on zoonotic arbovirus transmission

Tijani A. Sulaimon[1,2,3]*, Anthony J. Wood[2], Michael B. Bonsall[4], Michael Boots[5], Jennifer S. Lord[1]

1 Department of Vector Biology, Liverpool School of Tropical Medicine, Liverpool, United Kingdom, 2 The Roslin Institute, University of Edinburgh, Easter Bush Campus, Midlothian, United Kingdom, 3 African Institute for Mathematical Sciences (AIMS), Muizenberg, Cape Town, South Africa, 4 Department of Biology, University of Oxford, Oxford, United Kingdom, 5 Department of Integrative Biology, University of California, Berkeley, California, United States of America

* tijani@aims.ac.za

**Data availability statement:** No new data were created in this study. Data sharing is not

## Abstract

The interaction between vector host preference and host availability on vector blood feeding behaviour has important implications for the transmission of vector-borne pathogens. However, to our knowledge, the effect of bias towards feeding on the same host species from which a first meal was taken, termed fidelity, has not been quantified. Using a mathematical model we showed that vector fidelity to the host species they take a first blood meal from leads to non-homogeneous mixing between hosts and vectors. Taking Japanese encephalitis virus (JEV) as a case study, we investigated how vector preference for amplifying vs dead-end hosts and fidelity can influence JEV transmission. We show that in regions where pigs (amplifying hosts) are scarce compared to cattle (dead-end hosts preferred by common JEV vectors), JEV could still be maintained through vector fidelity. Our findings demonstrate the importance of considering fidelity as a potential driver of transmission, particularly in scenarios such as Bangladesh and India where the composition of the host community might initially suggest that transmission is not possible.

## Author summary

In a system with different types of animals, the spread of mosquito-borne pathogens, such as Japanese encephalitis virus (JEV), depends on the animals mosquitoes choose to feed on. A behaviour called "fidelity", where mosquitoes tend to feed on the same type of animal after their first blood meal, can influence how pathogens spread. Using a mathematical model, we explored how this behaviour affects the transmission of JEV, a pathogen that relies on pigs to amplify the virus, while some other animals, like cattle, serve as dead-end hosts that do not contribute to its spread. Our results show that

applicable to this article. Analysis code is available at https://github.com/tijanisulaimon/mosquito-fidelity-model.

**Funding:** JSL is funded by a MRC Career Development Award MR/W017059/1.

mosquito fidelity can sustain JEV transmission even when pigs are scarce compared to cattle. This behaviour could create localised transmission cycles when fidelity is sufficiently high, as mosquitoes feeding on pigs are more likely to return to them for future meals.

## Introduction

It is estimated that there are > 14 000 species of hematophagous arthropods [1], some of which are vectors of vertebrate pathogens. Arthropod vectors have evolved, to a greater or lesser degree, particular preferences for certain host species, or groups of hosts, from which to take a blood meal [2,3]. Here we define host preference as the genetically based tendency to respond to particular host cues, which influences how a mosquito chooses a host species to feed on for their first blood feed [4]. During host-seeking, genetically based host preference combines primarily with relative host availability and vector density, to determine realised host choice. For example, [5] showed that triatomines fed less frequently on humans when dogs or chickens were present in bedroom areas and when triatomine density was higher. Similarly, Thiemann et al. [6] demonstrated that temporal changes in the blood feeding patterns of the mosquito *Culex tarsalis*, from ardeid birds in the summer to mammals once ardeids had fledged and left the area, were driven by changes in host availability and mosquito abundance in addition to underlying preference.

The interaction between vector host preference and host availability on vector blood feeding behaviour has consequences for the transmission dynamics of vector-borne pathogens. Miller and Huppert [7], using a general modelling framework, showed that higher host diversity can result in amplification of a pathogen when the preference for highly competent host species is sufficiently strong to overcome any losses in transmission to less-competent species. However, diversity can cause dilution when a vector prefers host species that have lower competence, or when a vector does not have an underlying preference for any specific host species. Researchers have also carried out analyses for specific pathogens and contexts with respect to the effect of host choice. Killeen et al. [8] used models parameterised by field data to show that increased cattle populations could reduce malaria transmission in The Gambia but not in Tanzania due to higher *Anopheles* spp. preference for cattle compared with humans in The Gambia. Similarly, Hamer et al. [9] showed that spatial variation in West Nile virus transmission can in part be explained by variation in vector host choice.

In most analyses, it is assumed that the relationship between vector host choice and relative host availability is linear. However, Yakob and Walker [10] modelled different functional responses of vectors to relative host availability and demonstrated that the assumed function can have implications for predicted effects of control measures aimed at reducing vector biting rates. Kingsolver [11] also used models to show that vector preference for infected hosts relative to susceptible hosts can have important effects on disease dynamics. Despite this range of modelling efforts, to our knowledge, there have been no analyses of the possible effects of vector memory and host fidelity on host choice and pathogen transmission dynamics. Fidelity refers to the bias towards feeding on the same host type for subsequent feeds after the first blood feed.

There is empirical evidence that vectors can learn from previous experience [12]. For mosquitoes, this is both with respect to choice of oviposition sites and hosts. There is evidence that both attraction and aversion behaviour can be learned when mosquitoes are exposed to a particular odour alongside either a positive or negative experience [13,14]. We are not aware of any modelling studies that consider the potential effects of vector fidelity on the

transmission dynamics of zoonotic vector-borne pathogens. Given that, by necessity, the community of vector species that transmit zoonotic pathogens feed on multiple host species, the question of the effect of learning during host choice on their transmission dynamics is particularly pertinent.

Our main goal was to fill this research gap by introducing a model where vectors become "imprinted" to a host species on their first blood meal and have a degree of fidelity to them for future feeds. This extends the traditional compartmental model [15,16] to have a "memory" parameter. We use this to assess the impact of fidelity on transmission dynamics between vector and host using analytic and numerical methods. We use Japanese encephalitis virus (JEV) as a case study for this model. Japanese encephalitis is a zoonotic disease caused by JEV [17], primarily transmitted by mosquitoes [18,19]. It poses a significant health threat, particularly in rural areas of Asia where rice cultivation and pig farming are prevalent [20]. There is no specific treatment available for JE, but the human disease is preventable by vaccination [21].

The transmission of JEV involves a cycle that occurs between multiple species of mosquitoes and reservoir hosts, including pigs and birds [18,20,22]. To maintain the enzootic cycle of JEV, a mosquito must become infected by biting an infectious host and subsequently survive, develop a sufficient viral load in the saliva for transmission, find, and bite a susceptible host. For an infected host to transmit the virus to a mosquito vector, it must attain an adequate concentration of the virus in its bloodstream. Most importantly, two studies have provided evidence for host fidelity in potential vectors of JEV. Mwandawiro et al. [23,24] showed through field experiments that although *Culex vishnui*, *Culex tritaeniorhynchus* and *Culex gelidus* would more frequently choose a cow over a pig in host choice experiments, they had a tendency to take a subsequent blood meal on the host species from which they had previously fed. Those initially attracted to pigs were more likely to take subsequent blood meals from pigs.

Intensive pig farming is associated with the prevalence of JE in regions such as China and Japan [20,25]. Pigs are the main hosts for JEV amplification and may re-transmit to susceptible feeding mosquitoes, whereas evidence suggests cattle are dead-end hosts and do not contribute to JEV transmission [26]. Hence, high relative availability of cattle compared with pigs may dilute JEV transmission due to the preference of JEV vectors for this host. However, JE cases can still occur in regions with low pig density compared to cattle, such as certain areas of Bangladesh and India [25]. For example, in three JE-endemic districts of Rajshahi Division, Bangladesh, the ratio of cattle to pigs is approximately 140:1. Similarly, in some JE-endemic regions of India, cattle can outnumber pigs by a ratio of up to 20:1. In these regions where there is a relatively high density of cattle, which are preferred by *Culex tritaeniorhynchus* mosquitoes, could the fidelity behaviour of mosquitoes explain the persistence of JEV?

## Materials and methods

### A vector-borne pathogen model with host preference and fidelity

We describe a model of pathogen transmission that involves two host species and a vector. This is a general compartmental model, though in this work we consider model parameters so as to capture the dynamics of JEV over host populations of pigs and cows and a mosquito vector population.

**Fidelity-free transmission dynamics.**   To begin, the host population comprises two distinct groups; *amplifying* hosts ($A$) and *dead-end* hosts ($D$). The amplifying host population ($H_A$) is capable of replicating and transmitting the virus to vectors. The dead-end host population ($H_D$) can be infected but does not contribute to transmission. For each host species $i \in \{A, D\}$, we categorise into three compartments: susceptible ($S_i$), infected ($I_i$), and recovered

($R_i$) individuals, with the total population given by $H_i = S_i + I_i + R_i$. We assume the infection is nonfatal and all infected hosts recover at rate $\gamma$. For simplicity, the birth and death rates of both hosts are fixed at $\mu$, preserving the total population in time.

The mosquito population follows a susceptible-infected dynamic (SI). Mosquitoes are introduced at a rate $\mu_m$ into the susceptible population $S^m$. Mosquitoes die at rate $\mu_m$, preserving the total population, and bite hosts at a rate $\alpha$. The mosquito's choice of host type is determined (noting we are not yet considering fidelity) based on a host preference. We define the probability $\rho_i$ that a mosquito chooses host species $i$ for a particular blood meal as:

$$\rho_i = \frac{p_i H_i}{\sum_{\text{all host species } j} p_j H_j},$$
(1)

with $p_i$ the genetically based host preference for species $i$ and $j$ represents each individual host species.

If a susceptible mosquito bites an infected amplifying host ($I_A$), that mosquito becomes infected with probability $\beta$. Similarly, if an infected mosquito ($I^m$) bites a susceptible host (of any species), that host becomes infected with probability $\beta$. Susceptible mosquitoes, therefore, enter the infected compartment at rate $\rho_A \alpha \beta S^m \frac{I_A}{H_A}$. Hosts $i$ enter the infected compartment at rate $\rho_i \alpha \beta I^m \frac{S_i}{H_i}$. The definitions of the parameters used in the model are presented in Table 1.

Combining these infection dynamics, host recovery and the birth-death processes, the time evolution of this process can be described by the following set of ordinary differential equations:

**Mosquitoes**

$$\frac{S^m}{dt} = \mu_m N^m - \rho_A \alpha \beta S^m \frac{I_A}{H_A} - \mu_m S^m$$

$$\frac{I^m}{dt} = \rho_A \alpha \beta S^m \frac{I_A}{H_A} - \mu_m I^m$$

**Amplifying host**

$$\frac{S_A}{dt} = \mu H_A - \alpha \beta \rho_A I^m \frac{S_A}{H_A} - \mu S_A$$

$$\frac{I_A}{dt} = \alpha \beta \rho_A I^m \frac{S_A}{H_A} - \gamma I_A - \mu I_A$$

$$\frac{R_A}{dt} = \gamma I_A - \mu R_A$$

**Dead-end host**

$$\frac{S_D}{dt} = \mu H_D - \alpha \beta \rho_A I^m \frac{S_D}{H_D} - \mu S_D$$

$$\frac{I_D}{dt} = \alpha \beta \rho_A I^m \frac{S_D}{H_D} - \gamma I_D - \mu I_D$$

$$\frac{R_D}{dt} = \gamma I_D - \mu R_D$$

(2)

**Incorporating mosquito fidelity behaviour.** The model described in Eq. 2 assumes that the host choice of a mosquito depends only on the relative host abundance, and the

**Table 1. Definition of parameters used in the model.**

| Symbol | Meaning |
|---|---|
| $H_A$ | Total number of individuals of amplifying host species (constant, $H_A = S_A + I_A + R_A$) [individual per area] |
| $H_D$ | Total number of individuals from dead-end host species (constant) [individual per area] |
| $p_i$ | Genetically based host preference for species $i$ [unitless] |
| $\alpha$ | Mosquito biting rate [time$^{-1}$] |
| $\beta$ | Transmission probability: mosquito to competent host and vice versa [unitless] |
| $\gamma$ | Recovery rate for host species (inverse of infectious duration) [time$^{-1}$] |
| $\mu_m$ | Per capita background mosquito mortality and birth rate (inverse of mosquito life expectancy) [time$^{-1}$] |
| $\mu$ | Per capita background host mortality and birth rate [time$^{-1}$] |
| $f$ | Mosquito fidelity (only bites imprinted host species if $f$ = 1; bites host species $i$ with probability $\rho_i$ if $f$ = 0) [unitless] |

mosquito's *genetically based preference* (Eq. 1). We now extend this to include a *fidelity* dynamic. For a mosquito imprinted on host type $i$, the mosquito remains "loyal" to $i$ for any subsequent bite with probability $f \in [0, 1]$. If the mosquito does not remain loyal (probability $1-f$), then its choice of host is determined by the preference described in Eq. 1. When $f$ = 0 (no fidelity), mosquitoes behave solely based on host density and initial preference. Conversely, for $f$ = 1 (total fidelity), mosquitoes only bite the host type that its first bite was taken from (Fig 1). For subsequent bites by mosquitoes imprinted on host $i$, the proportion of bites on host $i$ becomes $\rho_i + f\rho_{\neg i}$, where $\rho_{\neg i}$ represents the proportion of first mosquito bites on the other host type. In contrast, for mosquitoes imprinted on the other host type, the proportion of subsequent bites on host $i$ is $(1 - f)\rho_i$.

To generalise Eq. 2 to explore fidelity, we need to label mosquitoes based on the host (if any) they are imprinted on to. We therefore label mosquitoes as either *nulliparous* (yet to take a blood meal) ($S^m$), imprinted on an amplifying host ($S_A^m$, or $I_A^m$ if infected), and imprinted on a dead-end host ($S_D^m$, $I_D^m$ if infected), bringing a total of five compartments. This allows us to keep track of mosquito activity once they have imprinted, for example, the compartment $I_D^m$ allows for mosquitoes that initially imprinted on a cow, but by probability $(1 - f)\rho_A$, may feed on a pig and potentially become infected, however they still have probability $[f + (1 - f)\rho_D]$ of feeding on a cow subsequently.

Susceptible mosquitoes imprinted on amplifying and dead-end hosts are recruited from the nulliparous group at rates $\rho_A \alpha \left(1 - \beta \frac{I_A}{H_A}\right)$ and $\rho_D \alpha$, respectively, where $\rho_A \alpha \beta \frac{I_A}{H_A}$ is the rate at which nulliparous mosquitoes become infected by taking their first blood meal on infected amplifying hosts. The rate at which susceptible mosquitoes imprinted on a host species $A$ become infected is $[f + (1 - f)\rho_A] \alpha \beta \frac{I_A}{H_A}$. Incorporating this subclassification of mosquitoes, Eq. 2 generalises to:

## (a) Biting behaviour

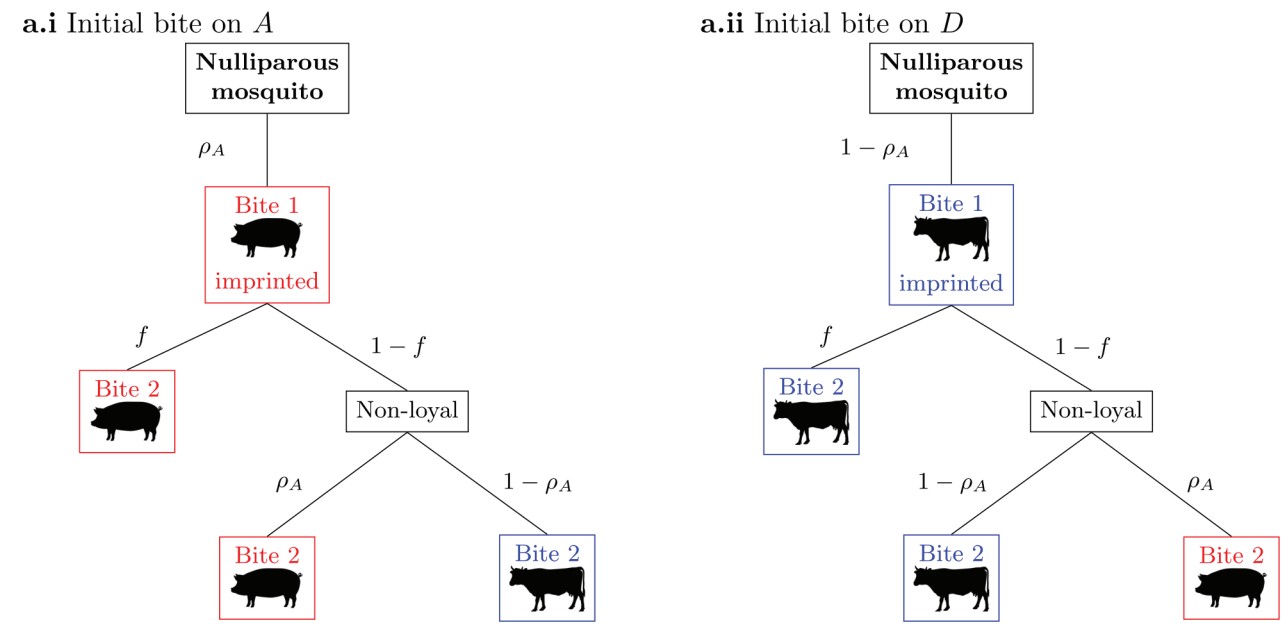

## (b) Conceptual models of Japanese encephalitis virus (JEV) transmission

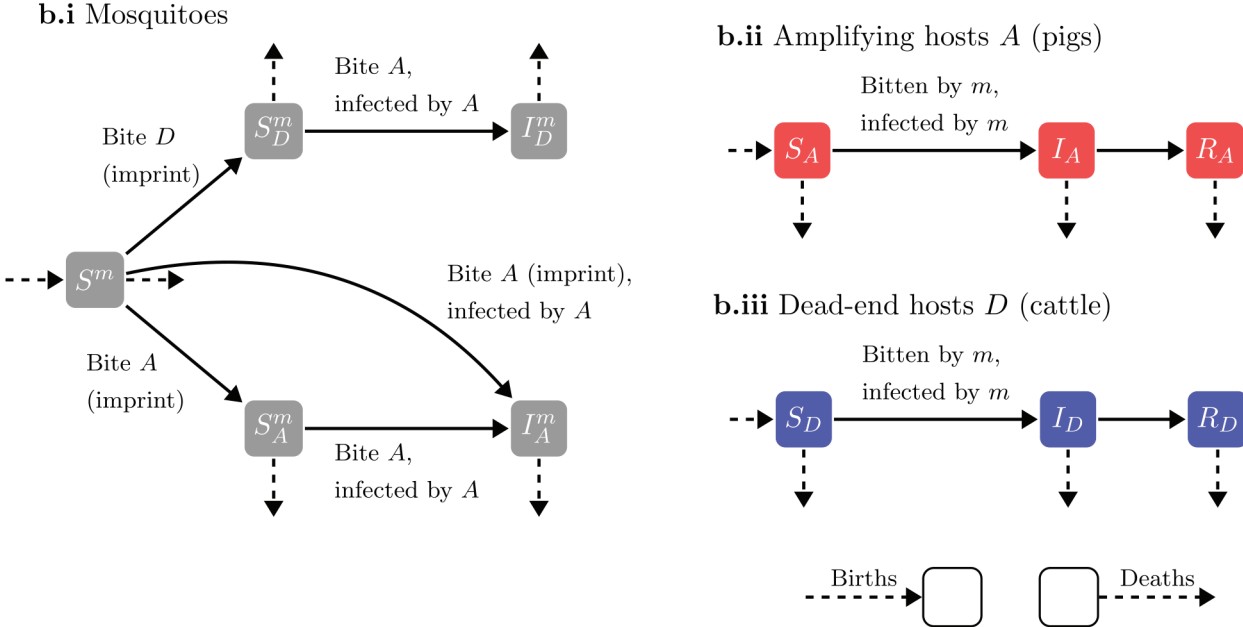

**Fig 1. (a): Biting behaviour of mosquitoes, when (i) imprinted on an amplifying host $A$, and (ii) on a dead-end host $D$.** A mosquito's first bite is determined by host density and the host preference (modulated by the parameters $\rho_A$, $\rho_D = 1 - \rho_A$). All subsequent bites are determined by a mixture of fidelity (modulated by the parameter $f$), and host preference if the mosquito is not loyal to its imprinted host species. **(b)**: Compartmental models. (i) Mosquitoes are introduced nulliparous ($S^m$), and imprinted onto the host species they first bite ($S^m \rightarrow S_D^m$, $S^m \rightarrow S_A^m$), and later may become infected by biting infected amplifying hosts ($S_D^m \rightarrow I_D^m$, $S_A^m \rightarrow I_A^m$, with potential for $S^m \rightarrow I_A^m$ if infected on their imprinting bite). Mosquitoes can die in any compartment. (ii), (iii) Amplifying and dead-end hosts respectively. Hosts can become infected if bitten by an infected mosquito and later go on to recover. Hosts are born susceptible and can die in any compartment. Cow and pig images obtained from https://www.phylopic.org, licensed under CC0 1.0.

**Mosquitoes**

$$\frac{S^m}{dt} = -\mu_m S^m + \mu_m N^m - \alpha S^m$$

$$\frac{S_A^m}{dt} = -\mu_m S_A^m - \left[f + (1-f)\rho_A\right]\alpha\beta\frac{I_A}{H_A}S_A^m + \rho_A\alpha\left(1 - \beta\frac{I_A}{H_A}\right)S^m$$

$$\frac{I_A^m}{dt} = -\mu_m I_A^m + \left[f + (1-f)\rho_A\right]\alpha\beta\frac{I_A}{H_A}S_A^m + \rho_A\alpha\beta\frac{I_A}{H_A}S^m$$

$$\frac{S_D^m}{dt} = -\mu_m S_D^m - \left[(1-f)\rho_A\right]\alpha\beta\frac{I_A}{H_A}S_D^m + (1-\rho_A)\alpha S^m$$

$$\frac{I_D^m}{dt} = -\mu_m I_D^m + \left[(1-f)\rho_A\right]\alpha\beta\frac{I_A}{H_A}S_D^m$$

**Amplifying hosts**

$$\frac{S_A}{dt} = -\mu S_A - \alpha\beta\left[f + (1-f)\rho_A\right]I_A^m\frac{S_A}{H_A} - \alpha\beta\left[(1-f)\rho_A\right]I_D^m\frac{S_A}{H_A} + \mu H_A$$

$$\frac{I_A}{dt} = -\mu I_A + \alpha\beta\left[f + (1-f)\rho_A\right]I_A^m\frac{S_A}{H_A} + \alpha\beta\left[(1-f)\rho_A\right]I_D^m\frac{S_A}{H_A} - \gamma I_A \qquad (3)$$

$$\frac{R_A}{dt} = -\mu R_A + \gamma I_A$$

**Dead-end hosts**

$$\frac{S_D}{dt} = -\mu S_D - \alpha\beta\left[f + (1-f)(1-\rho_A)\right]I_D^m\frac{S_D}{H_D} - \alpha\beta\left[(1-f)(1-\rho_A)\right]I_A^m\frac{S_D}{H_D} + \mu H_D$$

$$\frac{I_D}{dt} = -\mu I_D + \alpha\beta\left[f + (1-f)(1-\rho_A)\right]I_D^m\frac{S_D}{H_D} + \alpha\beta\left[(1-f)(1-\rho_A)\right]I_A^m\frac{S_D}{H_D} - \gamma I_D$$

$$\frac{R_D}{dt} = -\mu R_D + \gamma I_D$$

We first illustrate the implications of fidelity as described in Fig 1 on the host species choice of mosquitoes. The host population consists of an *equal* proportion of cows (dead-end hosts, *D*) and pigs (amplifying hosts, *A*). From the full model in Eq. 3, we derived an expression for the basic reproduction number ($R_0$) in the case of an infection-free equilibrium (see Sect A.1 in S1 Text), where populations are entirely susceptible. The basic reproduction number is a fundamental epidemiological metric that is used to assess the potential for pathogen transmission in a population. It represents the average number of secondary infections caused by a single infected individual in a fully susceptible population. Using the parameters described in Table 2, we analysed the influence of fidelity, host composition, and mosquito-to-host ratio on $R_0$.

We used experiments conducted by Mwandawiro et al. [24] to estimate the initial preference for amplifying host for the three mosquito species competent for JEV: *Cx. tritaeniorhynchus* ($\rho_A \approx 0.05$), *Cx. gelidus* ($\rho_A \approx 0.22$), and *Cx. vishnui* ($\rho_A \approx 0.15$). Subsequently, we used the mosquito biting behaviour described in Fig 1 to estimate the fidelity values of each mosquito species using maximum likelihood estimation (see Sect A.2 and Fig B in S1 Text).

## Results

### Host choice experiment

The implications of fidelity on the host species choice of mosquitoes are illustrated in Fig 2, which shows the proportion of *initial* feeds on the two host species and how the proportion of all *subsequent* feeds changes with fidelity. For no fidelity ($f = 0$, top row) where mosquitoes have no memory of the host they first fed on, the proportion of subsequent bites mirrors that of initial host preference. As the value for $f$ increases, feeds are increasingly from mosquitoes imprinted onto that species, where with total fidelity ($f = 1$, bottom row) mosquitoes exclusively feed on the host to which they were imprinted. Note that the proportion of subsequent feeds per host remains constant.

### Basic reproduction number

When there is a mixture of amplifying and dead-end hosts in the population, $R_0$ increases with an increasing initial preference for amplifying hosts, fidelity, and mosquito-to-host ratio (Fig 3). When preference for the amplifying host species is 0.01, and given the fixed parameter values (Table 2), $R_0$ is less than one when there are dead-end hosts in the population, except when the proportion of amplifying hosts $N_A/N$ exceeds 0.8 and the fidelity is sufficiently high ($f > 0.6$). With 50 mosquitoes per host and an amplifying host preference above 0.05, $R_0$ is greater than 1, regardless of fidelity. However, an epidemic will not spread ($R_0 < 1$) even if the mosquito-host ratio is as high as 50, provided that there are fewer amplifying hosts than dead-end hosts, the amplifying host preference less than 0.05 and $f$ is less than 0.2. In the absence of fidelity and with a very low preference for the amplifying host, $R_0$ is less than 1 if $N_A/N$ is less than 0.8, regardless of the number of mosquitoes per host. Therefore, fidelity can lead to $R_0 > 1$ in circumstances where it would otherwise expected to be <1.

### Transmission dynamics

Fig 4 shows the dynamics of infected host populations. In the absence of fidelity, an epidemic occurs only when there is an equal number of amplifying and dead-end hosts and the initial preference for the amplifying host is 0.2. In this scenario, the dead-end host population experiences a higher peak epidemic size than the amplifying host population, with the dead-end host reaching its peak first. Increasing fidelity results in larger peak epidemic sizes in both populations. However, beyond a certain level of fidelity, the peak epidemic size becomes larger in amplifying host populations than in dead-end populations. Perfect fidelity prevents epidemics in dead-end hosts, as those initially feeding on amplifying hosts maintain their preference.

**Table 2. Estimates and range of parameter values used in model analysis and global sensitivity analysis.**

| Parameter | Value | Range | Reference |
|---|---|---|---|
| $H$ | 1000 | (100,1000) | Assumed |
| $\rho_A$ | — | (0.01,0.5) | [23,24] |
| $\beta$ | 1 | (0,1) | Assumed |
| $\alpha$ | 1/3 day$^{-1}$ | (1/6, 1/2) | [27] |
| $\gamma$ | 1/5 day$^{-1}$ | (1/2, 1/7) | [26] |
| $\mu_m$ | 1/30 day$^{-1}$ | (1/35, 1/15) | Assumed |
| $\mu$ | 1/365 day$^{-1}$ | (1/1095, 1/90) | Assumed |

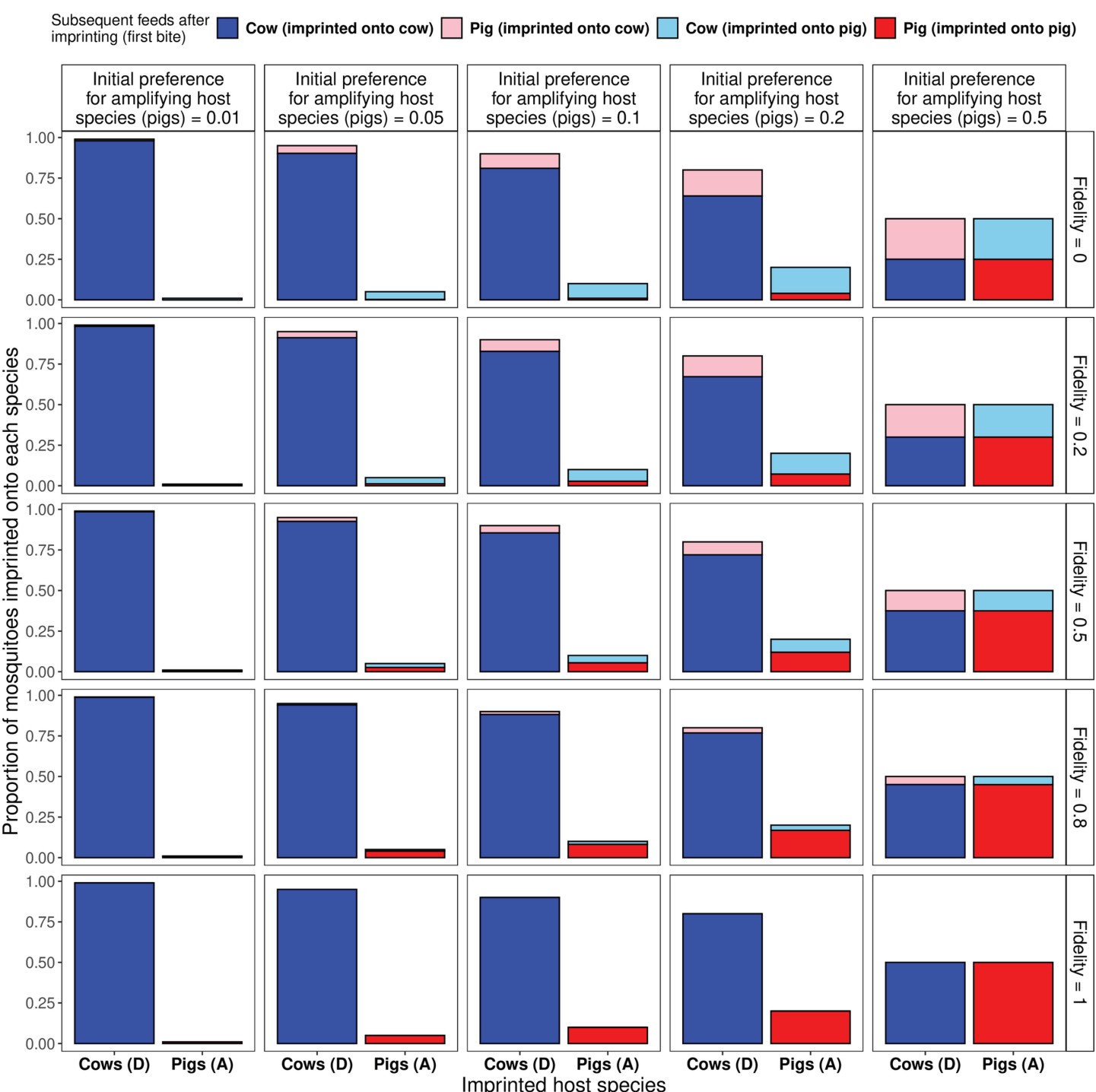

**Fig 2. Variation of feeding patterns of mosquitoes with fidelity, in the instance where cows (D) and pigs (A) are available in equal proportions.** The height of the bars indicates the proportion of mosquitoes that are imprinted on that host species (x-axis). The fill indicates the proportion of *subsequent bites* by those imprinted mosquitoes. With an increasing preference for pigs (left-to-right), a higher proportion of imprinting bites are on pigs. With increasing fidelity (top-to-bottom), the proportion of subsequent bites remaining loyal to the imprinted host increases. In the case of no fidelity ($f = 0$, top row), the proportion of subsequent bites replicates the proportion of initial bites. While the overall proportions feeding on D or A does not change with fidelity — as fidelity increases the population feeding on cattle vs pigs becomes increasingly isolated from each other.

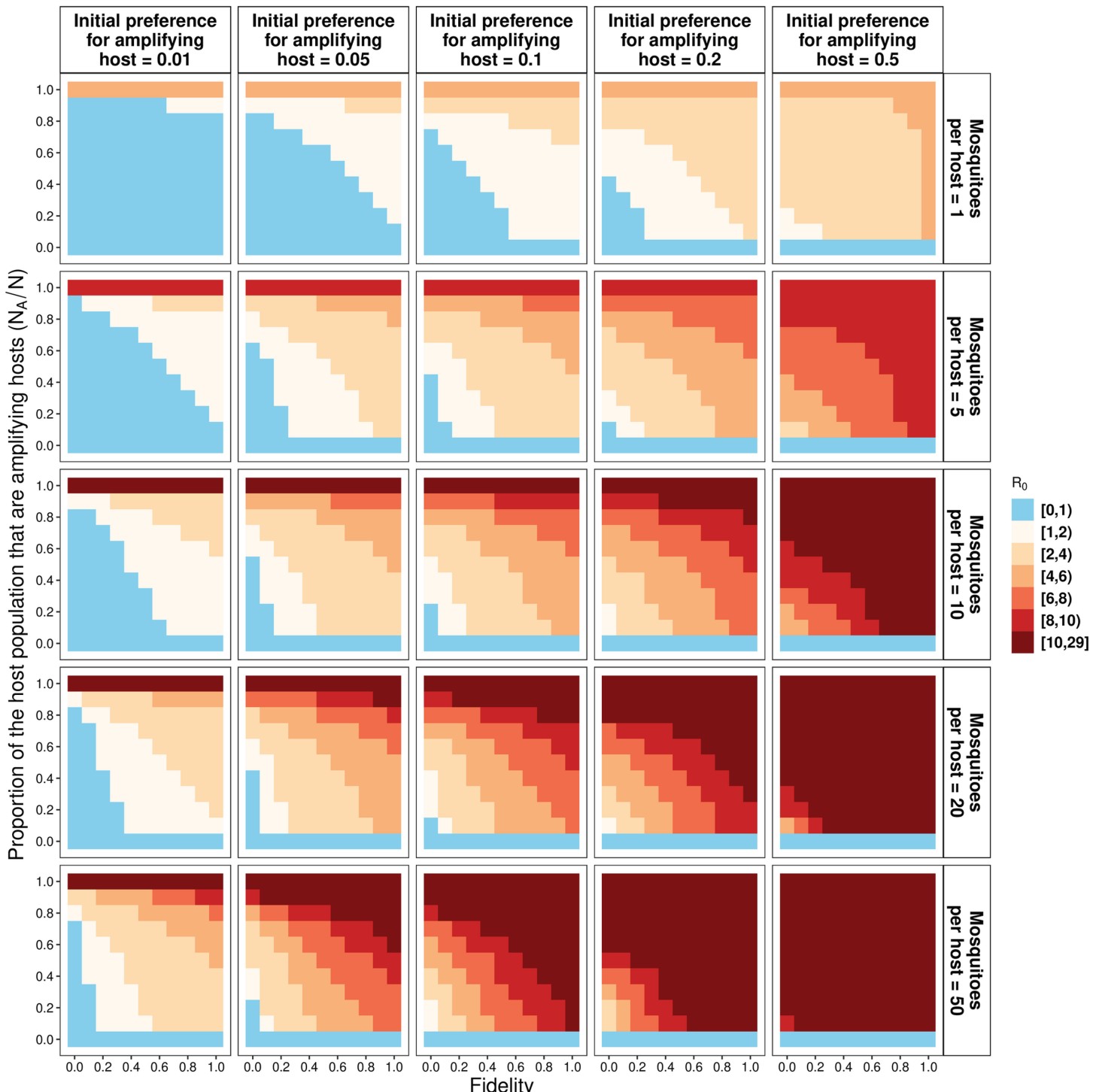

**Fig 3. Basic reproduction number $R_0$ values based on Eq B in S1 Text and values in Table 2.** The top panel represents the initial preference for amplifying hosts, and the right panel is the mosquito-to-host ratio. Blue indicates regions where $R_0 < 1$, i.e. there is no outbreak.

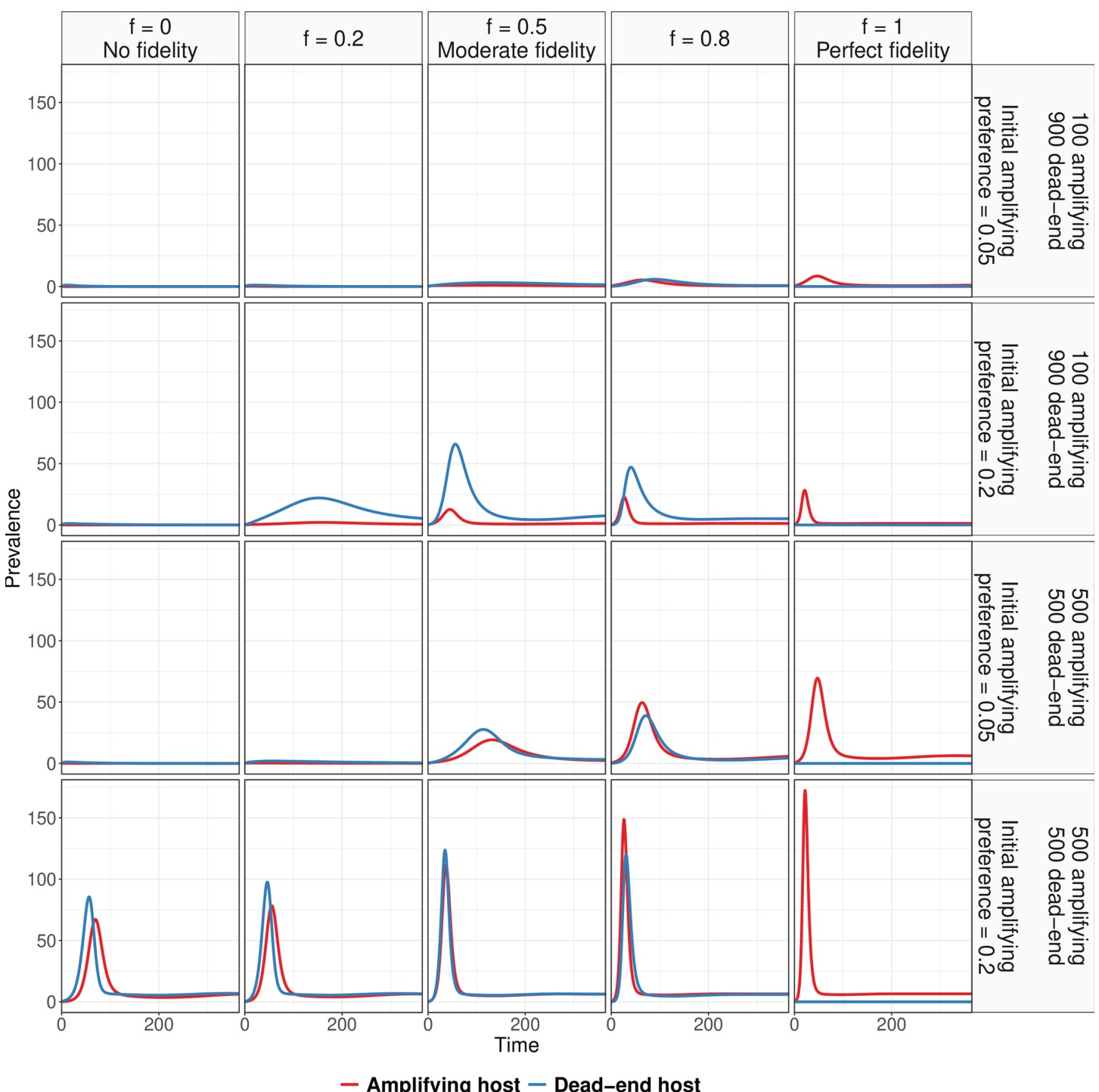

**Fig 4. Dynamics of infected amplifying and dead-end host assuming 5 mosquitoes per host and fixed parameter values in Table 2.** The right panel indicates host composition and initial preference for amplifying hosts, while the top panel indicates different fidelity values. While outbreak size in amplifying hosts increases with fidelity, less than perfect fidelity leads to larger outbreaks in the dead-end host.

In the amplifying host population, the epidemic size increases with increasing fidelity. However, in the dead-end population, the impact of fidelity varies based on initial preference and host composition. For example, when the amplifying to dead-end host ratio is 1, the peak epidemic size increases with increasing fidelity and initial preference. However, when there are 100 amplifying hosts and 900 dead-end hosts, and the initial preference is 0.2, the peak epidemic size increases with increasing fidelity values up to moderate fidelity, beyond which it decreases again. Cumulative incidence values under these different scenarios are provided in Table B in S1 Text.

### Comparison of relative $R_0$ between different mosquito species

The fidelity estimates and their 95% confidence interval are 0.53 $[0.31, 0.74]$ for *Cx. tritaeniorhynchus*, 0.57 $[0.26, 0.88]$ for *Cx. gelidus*, and 0.75 $[0.49, 1.00]$) for *Cx. vishnui*. Each of the three JEV vectors has the potential to trigger an outbreak in a population consisting of only 1% amplifying host, with an average of 5 mosquitoes per host (Fig 5). Note that the value of $R_0$ will depend on the extrinsic incubation period of mosquitoes, which is not considered here. In addition, other parameters used in this analysis are held constant under relatively lenient conditions (see caption in Fig 5). However, we performed a global sensitivity analysis to assess the variability of $R_0$ across all parameter ranges, as shown in Fig C in S1 Text.

*Cx. gelidus* has an initial preference of 0.22 for amplifying host species, while *Cx. tritaeniorhynchus* has an initial preference of 0.05. *Cx. gelidus* has a higher initial preference for amplifying host species compared to *Cx. tritaeniorhynchus*. Despite similar fidelity estimates, the potential for an outbreak is substantially higher in *Cx. gelidus*. On the other hand, for *Cx. gelidus* and *Cx. vishnui* with similar initial preferences, the potential for an outbreak remains comparable, even though the fidelity estimate of *Cx. vishnui* is substantially higher than that of *Cx. gelidus*. These findings are assuming equal abundance of each species.

## Discussion

Here we have shown how vector memory of, and subsequent fidelity to, the host species they take a first blood meal from, can permit invasion and onward transmission of a zoonotic mosquito-borne virus in contexts where this would not be expected otherwise. This may include circumstances where vector species prefer dead-end hosts and where dead-end hosts are the most common vertebrate species. Although vector fidelity does not affect the overall proportion of blood meals on each host species, we have shown how it can lead to non-homogeneous mixing between the host and vector community; when a mosquito has fidelity for host species, those mosquitoes who first feed on amplifying hosts are more likely to take subsequent feeds from amplifying hosts.

In our analyses, we have used JEV as an example of the impact of fidelity on transmission. Our simulations demonstrate that in regions where pigs are rare relative to cattle, which are dead-end hosts but preferred by common JEV vectors, JEV could be maintained if vectors display sufficient host species fidelity. While we used the example of the role of pigs and cattle in JEV transmission, the findings may also extend to poultry. As highlighted by Lord et al. [25] poultry are common in many regions where JEV occurs and produce sufficient viremia to infect mosquitoes, but their role in transmission has not been fully quantified. The extent to which fidelity may contribute to the role of poultry in JEV transmission is unknown. It is likely that both presence and abundance of ornithophilic mosquitoes and fidelity in more generalist vectors will contribute to determining what role poultry play in JEV transmission in a given ecological context. These findings have implications for any risk assessment of JEV

spread to new areas that involves consideration of host and vector species community composition; transmission could be sustained in communities dominated by dead-end hosts because of vector host species fidelity.

There is empirical evidence that the fidelity of *Cx. tritaeniorhynchus* may be lower than that of *Cx. vishnui* and its preference for pigs may be lower than both *Cx. vishnui* and *Cx. gelidus* [24]. This has potential implications for the relative role of each vector species in JEV transmission in regions where cattle are the dominant vertebrate host; *Cx. tritaeniorhynchus* may not always be the most important vector. While our analysis has shown that, by incorporating fidelity, *Cx. tritaeniorhynchus* may be less important to transmission, we acknowledge this is when assuming that all vector species are present in equal abundance. In many

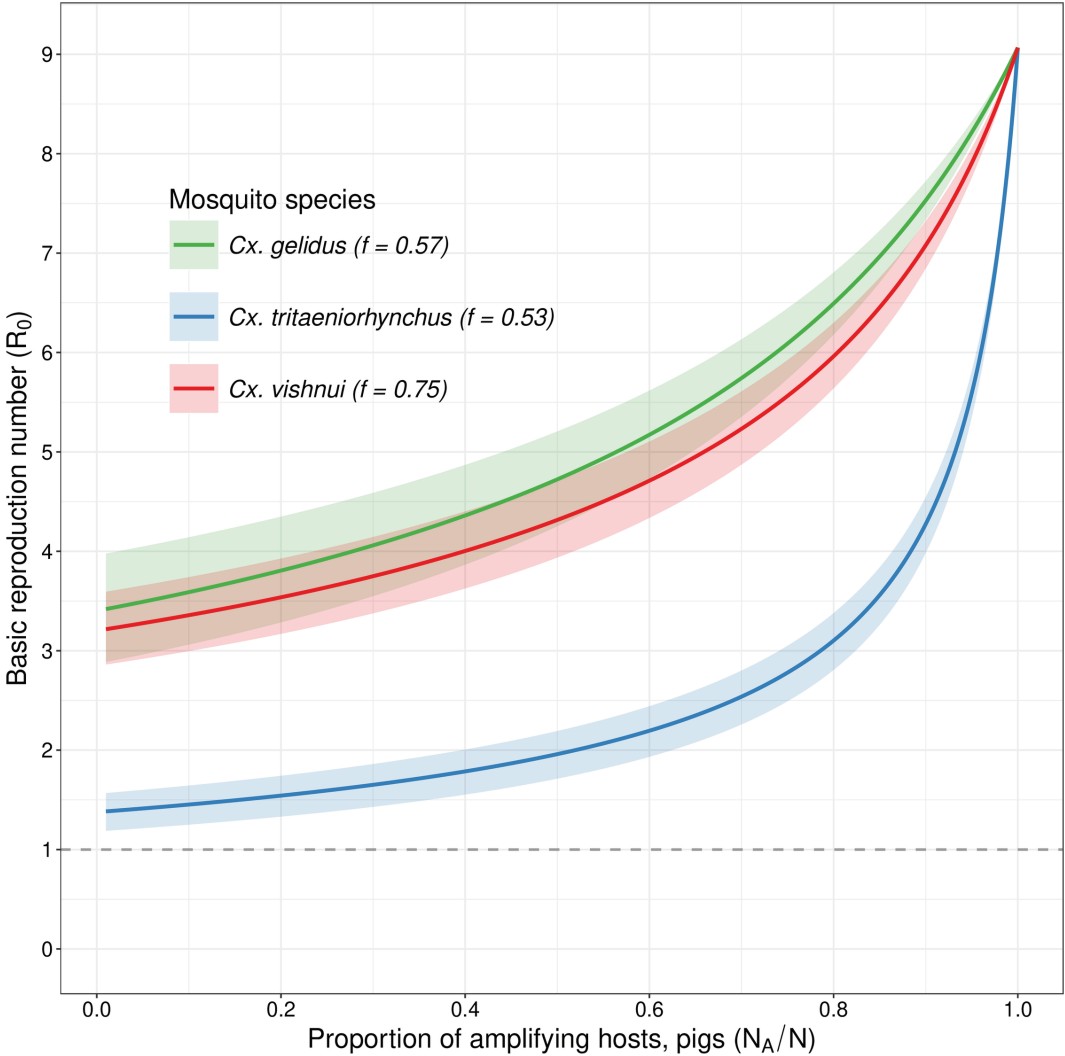

**Fig 5. Basic reproduction number** ($R_0$) **for Cx. tritaeniorhynchus, Cx. gelidus, and Cx. vishnui based on varying host composition** ($\frac{H_A}{H_D} \in [0.01, 1]$)**.** The initial preference for amplifying host species ($\rho_A$) and fidelity values ($f$) are provided in Table A in S1 Text, along with their corresponding 95% confidence intervals. Other parameters held constant: transmission probability—1, biting rate—1/3 day$^{-1}$, mosquito mortality and birth rate—1/30 day$^{-1}$, host recovery rate—1/5 day$^{-1}$, host mortality, birth rate—1/365 day$^{-1}$, and mosquitoes-to-host ratio—5.

times and places, *Cx. tritaeniorhynchus* may be substantially more abundant than other vector species and thus still be the most important vector, despite lower levels of fidelity to amplifying hosts [25]. More generally, it may be important to consider fidelity as well as vector abundance, competence and host preference when incriminating mosquito species in pathogen transmission.

Vector fidelity may be important to consider for other vector-borne disease systems. For example, there are contexts where zoophilic vectors are important in malaria transmission [28–30] and the role of cattle in transmission and control of malaria has been assessed. The presence of cattle can either exacerbate (zoopotentiation) or reduce transmission (zooprophylaxis), depending on the location of cattle with respect to humans and on vector feeding behaviour [31]. While in some circumstances the presence of cattle may increase the incidence of malaria, their role in control could be enhanced by the use of insecticides. Waite et al. [30] combined empirical studies with modelling to show that increasing vector mortality during feeding on non-human hosts can contribute to reducing malaria transmission. Ruiz-Castillo et al. [32] also highlight potential strategies including insecticide-treated cattle. However, we are not aware of any analyses of malaria control using livestock that factor in the potential role of vector fidelity. Non-homogeneous mixing caused by fidelity could undermine interventions aimed at livestock. However, modelling to further explore this would be necessary, in particular to quantify how fidelity may also cause increased vector density and subsequent effects of both altered mosquito to host ratio and host choice on transmission.

In our model, we considered the transmission of JEV within two host species, thereby neglecting the potential influence of other host species that could serve as alternative sources of blood meals. However, our model is adaptable and can be extended to include a broader range of alternative hosts. In addition, we ignored some ecological factors that could affect the potential of outbreaks, including the seasonal variation in host availability, which could directly impact on mosquito fidelity behaviour. While we have explored simulation scenarios for a broad range of host preference and fidelity, our analysis of species specific outbreak risk ($R_0$) relies on fidelity and host preference estimated from a single experimental study conducted by Mwandawiro et al. [24].

Our work contributes to the wider vector-borne pathogen modelling literature, which in the last decade has expanded beyond the original Ross-MacDonald framework. Obtaining species-level fidelity estimates for different vector species would require semi-field experiments and may often not be possible. In addition, there are also spatial heterogeneities, including in host community composition, that likely affect transmission. Here our point is to highlight the potential role of fidelity and for this to be considered as a potential driver of transmission in regions where host community composition would, in the first instance, suggest that transmission is not possible.

## Supporting information

**S1 Text. A**. Supplementary methods: A.1 Derivation of the basic reproduction number; A.2 Estimation of species-specific host preference and fidelity. **B**. Supplementary plots: mosquito feeding patterns under varying fidelity (Fig A), likelihood plot for fidelity estimation (Fig B), and global sensitivity analysis (Fig C). **C**. Supplementary tables: estimated host preference and fidelity values (Table A) and cumulative incidence of infected hosts (Table B).
(PDF)

## Acknowledgments

The authors gratefully acknowledge Juliet Pulliam, who originally conceptualised and developed the original version of the model with Mike Boots.

## Author contributions

**Conceptualization:** Tijani A. Sulaimon, Michael Boots.

**Formal analysis:** Tijani A. Sulaimon, Anthony J. Wood, Jennifer S. Lord.

**Investigation:** Tijani A. Sulaimon, Anthony J. Wood, Michael B. Bonsall, Michael Boots, Jennifer S. Lord.

**Methodology:** Tijani A. Sulaimon, Anthony J. Wood, Michael Boots, Jennifer S. Lord.

**Software:** Tijani A. Sulaimon, Anthony J. Wood, Jennifer S. Lord.

**Supervision:** Michael B. Bonsall, Jennifer S. Lord.

**Visualization:** Tijani A. Sulaimon, Anthony J. Wood, Jennifer S. Lord.

**Writing – original draft:** Tijani A. Sulaimon, Anthony J. Wood, Jennifer S. Lord.

**Writing – review & editing:** Tijani A. Sulaimon, Anthony J. Wood, Michael B. Bonsall, Michael Boots, Jennifer S. Lord.

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
