## [Decision Letter · Decision Letter 0]

13 Dec 2024

PNTD-D-24-00646The potential impacts of vector host species fidelity on zoonotic arbovirus transmissionPLOS Neglected Tropical Diseases  Dear Dr. Sulaimon,

Thank you for submitting your manuscript to PLOS Neglected Tropical Diseases. After careful consideration, we feel that it has merit but does not fully meet PLOS Neglected Tropical Diseases's publication criteria as it currently stands. Therefore, we invite you to submit a revised version of the manuscript that addresses the points raised during the review process.

Please submit your revised manuscript within 60 days Feb 11 2025 11:59PM. If you will need more time than this to complete your revisions, please reply to this message or contact the journal office at plosntds@plos.org. Please include the following items when submitting your revised manuscript: * A rebuttal letter that responds to each point raised by the editor and reviewer(s). You should upload this letter as a separate file labeled 'Response to Reviewers'. This file does not need to include responses to any formatting updates and technical items listed in the 'Journal Requirements' section below. * A marked-up copy of your manuscript that highlights changes made to the original version. You should upload this as a separate file labeled 'Revised Manuscript with Track Changes'. * An unmarked version of your revised paper without tracked changes. You should upload this as a separate file labeled 'Manuscript'. If you would like to make changes to your financial disclosure, competing interests statement, or data availability statement, please make these updates within the submission form at the time of resubmission. Guidelines for resubmitting your figure files are available below the reviewer comments at the end of this letter. We look forward to receiving your revised manuscript. Kind regards, Christopher M. BarkerAcademic EditorPLOS Neglected Tropical Diseases Amy GilbertSection EditorPLOS Neglected Tropical Diseases

Shaden Kamhawi

co-Editor-in-Chief

Paul Brindley

co-Editor-in-Chief

**Journal Requirements:**

At this stage, the following Authors/Authors require contributions: Tijani A Sulaimon, Anthony J Wood, Michael B Bonsall, Michael Boots, and Jennifer S Lord. Please ensure that the full contributions of each author are acknowledged in the "Add/Edit/Remove Authors" section of our submission form.

5) We have noticed that you have cited Table  Tables 1 and 2 not cited. in the manuscript file but there is no corresponding table in the manuscript.  Please amend your manuscript to include this table noting that tables should not be uploaded as individual files.

6) We have noticed that you have uploaded Supporting Information files, but you have not included a list of legends. Please add a full list of legends for your Supporting Information files after the references list.

7) We notice that your supplementary Figures are included in the manuscript file. Please remove them and upload them with the file type 'Supporting Information'. Please ensure that each Supporting Information file has a legend listed in the manuscript after the references list.

8) Some material included in your submission may be copyrighted. According to PLOSu2019s copyright policy, authors who use figures or other material (e.g., graphics, clipart, maps) from another author or copyright holder must demonstrate or obtain permission to publish this material under the Creative Commons Attribution 4.0 International (CC BY 4.0) License used by PLOS journals. Please closely review the details of PLOSu2019s copyright requirements here: PLOS Licenses and Copyright. If you need to request permissions from a copyright holder, you may use PLOS's Copyright Content Permission form.

Potential Copyright Issues:

- Figure 1. Please confirm whether you drew the images / clip-art within the figure panels by hand. If you did not draw the images, please provide a link to the source of the images or icons and their license / terms of use; or written permission from the copyright holder to publish the images or icons under our CC BY 4.0 license. Alternatively, you may replace the images with open source alternatives. See these open source resources you may use to replace images / clip-art:

9) We note that your Data Availability Statement is currently as follows: "No new data were created or analysed in this study. Data sharing is not applicable to this article.". Please confirm at this time whether or not your submission contains all raw data required to replicate the results of your study. Authors must share the “minimal data set” for their submission. PLOS defines the minimal data set to consist of the data required to replicate all study findings reported in the article, as well as related metadata and methods (https://journals.plos.org/plosone/s/data-availability#loc-minimal-data-set-definition).

- The points extracted from images for analysis..

**Additional Editor Comments:** The analyses presented in this manuscript depend on the code developed, and use of a private Github repository is inadequate to ensure availability and reproducibility of the findings. Please include the code as supplementary information with the manuscript or post the code on a public, permanent repository. In addition to the reviewers’ feedback, a PDF is attached with editorial comments.  **Reviewers' Comments:** Reviewer's Responses to Questions

**Key Review Criteria Required for Acceptance?**

**Methods**

-Are the objectives of the study clearly articulated with a clear testable hypothesis stated?

-Is the study design appropriate to address the stated objectives?

-Is the population clearly described and appropriate for the hypothesis being tested?

-Is the sample size sufficient to ensure adequate power to address the hypothesis being tested?

-Were correct statistical analysis used to support conclusions?

-Are there concerns about ethical or regulatory requirements being met?

Reviewer #1: The objectives of the study are clearly articulated and the hypothesis is clearly tested. The study design is appropriate to test the stated objectives. The research occurred in silico, with adequate sample sizes. Statistical analyses were appropriate. No concerns about ethical or regulatory requirements.

Reviewer #2: -Are the objectives of the study clearly articulated with a clear testable hypothesis stated?

Yes

-Is the study design appropriate to address the stated objectives?

I have a concern with the chosen compartmentalization of susceptible mosquitoes. I'm not sure why there are five categories of mosquitoes in the model - I think its more about tractability, but I'm having a hard time finding the biology in it (especially the infected mosquito fed on dead end host compartment). The nulliparous compartment makes sense, but then to me its not clear how things are moving through the system.

If a mosquito is only infected by the amplifying hosts (pigs), then I think the objective of fidelity would be to model individuals who take a first blood meal on a cow who then take a 1+n blood meal on a pig. The infected compartment would then also represent a mix of individuals who became infected on their first bite + those that eventually become infected on blood meal 1+ n. Once a mosquito is infected (either from the first or second blood meal) it would remain infected, correct?

Delta(nulliparous) = remains as written

Delta(susceptible but blood fed) = (proportion nulliparous fed on an uninfected pig) + (proportion nulliparous/parous fed on any cow) - (proportion susceptible but blood fed who then take a 1+n blood meal on an infected pig)

Delta(infected, blood fed) = (proportion nulliparous fed on an infected pig) + (proportion susceptible but blood fed who then take a 1+n blood meal on an infected pig)

To me, and based on my translation from Figure 1 (which clearly displays the individual-level process of blood feeding) to equation 3 (which presents a population level process), fidelity only weighs that 1+n blood meal.

Please clarify these concerns.

-Is the population clearly described and appropriate for the hypothesis being tested?

Yes

-Is the sample size sufficient to ensure adequate power to address the hypothesis being tested?

NA

-Were correct statistical analysis used to support conclusions?

NA

-Are there concerns about ethical or regulatory requirements being met?

No

**Results**

-Does the analysis presented match the analysis plan?

-Are the results clearly and completely presented?

-Are the figures (Tables, Images) of sufficient quality for clarity?

Reviewer #1: The analysis presented matches the analysis plan. The results are presented clearly, although in the general comments I have a minor suggestion to make them clearer. Figures and tables are generally clear, although I have one comment on a supplemental figure (see general comments section).

Reviewer #2: -Does the analysis presented match the analysis plan?

Yes

-Are the results clearly and completely presented?

Yes

-Are the figures (Tables, Images) of sufficient quality for clarity?

Yes

**Conclusions**

-Are the conclusions supported by the data presented?

-Are the limitations of analysis clearly described?

-Do the authors discuss how these data can be helpful to advance our understanding of the topic under study?

-Is public health relevance addressed?

Reviewer #1: The conclusions are supported by the data presented. The limitations are mostly described. The authors clearly explain how the simulation can be helpful to advance our understanding of vector-borne diseases.

Reviewer #2: -Are the conclusions supported by the data presented?

Yes

-Are the limitations of analysis clearly described?

Yes though the model also shows vector abundance can overcome limitations of preference/fidelity though this is not mentioned in the discussion or results.

-Do the authors discuss how these data can be helpful to advance our understanding of the topic under study?

Yes

-Is public health relevance addressed?

Yes

**Editorial and Data Presentation Modifications?**

Reviewer #1: See summary and General Comments below.

Reviewer #2: I found myself conflating definitions of preference, fidelity, (vector) memory, and imprinting. Preference is rightfully the genetic basis of host choice (the probability the mosquito a priori chooses a host). Fidelity - defined as imprinting, makes sense too bu that definition is presented too late in the manuscript. The distinction that fidelity only comes into play AFTER initial blood feeding could be a lot clearer in the report as well.

The introduction is a bit bumpy - meaning the flow of ideas sometimes is not as concise as it could be. For instance, over twenty lines are given to describing a single, prior published paper? Also, some paragraphs are only 1 - 2 sentences.

**Summary and General Comments**

Reviewer #1: The authors present a mathematical model to explore the role of host fidelity in mosquito-borne disease transmission, using Japanese encephalitis virus as an example. The article is well written. The conclusions follow logically from the model.

I have only minor comments:

The manuscript side-steps another potential mechanism for increasing amplification where amplifying hosts are less common and less preferred: local spatial heterogeneity. Even if pigs are outnumbered 140:1 at the regional scale, pigs may be locally concentrated and might outnumber cattle at the scale of an individual farm

Range for gamma in Table 3 is incorrect – should be fractions as for the other parameters.

The citation for the biting rate estimate was for Cx. quinquefasciatus. Is there a reason this species was chosen? I do not have access to the paper, to see if it provides biting rates for additional species. Taking a blood meal every 3 days strikes me as a fast turn-around for Culex, although biting rate is influenced by temperature. Not a major concern though, given the lack of sensitivity to biting rate in the sensitivity analysis.

Figure 4: A supplemental table with the total numbers infected would be helpful (or the numbers placed in the whitespace in the graph). I couldn’t quite tell which resulted in more infected overall – a longer lower peak or the sharper higher peaks.

Figure S2: Is the y-axis label correct? Isn’t this something to do with the likelihood of a given fidelity value? The methods used to calculate the fidelity estimates also come across as somewhat opaque to me – I’m not following the methods used here, except in the broadest sense.

Reviewer #2: In general, the manuscript investigates how blood meal imprinting (i.e., learned blood feeding) alters JEV transmission dynamics and then estimates vectoring potential of three important vector species. This is an interesting topic and relevant to the field of JEV and other VBD systems/models.

My primary scientific concern is that the fidelity-based ODEs, to me, don't match the biologically relevant stages of vectors - fidelity as presented in the manuscript should be more about tracking the development of infections in those species that initially blood fed on a dead end host, and simpler Susceptible/Infected classes of vectors may be just as efficient (and realistic) as those developed by the authors. My primary editorial concern is a stronger, clearer definition of fidelity is needed sooner in the manuscript and language needs to be more consistent between preference, fidelity, and imprinting.

A few other minor and specific comments:

Line 37: only anopheles (the genus) is listed (?)

Lines 174 - 183 should come sooner in the methods

Table 3: Mwandawiro 1999/2000 are foundational papers for this model. What were their estimates of fidelity - i.e., whats a realistic range of fidelity?

Lines 262 - 268: no citations?

Discussion of cattle and malaria extensions - I think an important caveat is that feeding on cattle (in amplification outcomes) generates increased vector abundance such that while the fraction of the infected population is small, the overall population is enormous!. In this instance, fidelity would lead to perhaps non-linear increases in vector abundance (and a secondary outcome would be nonlinear changes in transmission)

PLOS authors have the option to publish the peer review history of their article (what does this mean?). If published, this will include your full peer review and any attached files.

Reviewer #1: No

Reviewer #2: No

**Figure resubmission:** While revising your submission, please upload your figure files to the Preflight Analysis and Conversion Engine (PACE) digital diagnostic tool, https://pacev2.apexcovantage.com/. PACE helps ensure that figures meet PLOS requirements. To use PACE, you must first register as a user. Registration is free. Then, login and navigate to the UPLOAD tab, where you will find detailed instructions on how to use the tool. If you encounter any issues or have any questions when using PACE, please email PLOS at figures@plos.org. Please note that Supporting Information files do not need this step. If there are other versions of figure files still present in your submission file inventory at resubmission, please replace them with the PACE-processed versions.**Reproducibility:** To enhance the reproducibility of your results, we recommend that authors of applicable studies deposit laboratory protocols in protocols.io, where a protocol can be assigned its own identifier (DOI) such that it can be cited independently in the future. Additionally, PLOS ONE offers an option to publish peer-reviewed clinical study protocols. Read more information on sharing protocols at https://plos.org/protocols?utm_medium=editorial-email&utm_source=authorletters&utm_campaign=protocols

---

## [Decision Letter · Decision Letter 1]

2 Apr 2025

Dear Sulaimon,

We are pleased to inform you that your manuscript 'The potential impacts of vector host species fidelity on zoonotic arbovirus transmission' has been provisionally accepted for publication in PLOS Neglected Tropical Diseases.

Best regards,

Paul O. Mireji, PhD

Section Editor

Paul Mireji

Section Editor

Shaden Kamhawi

co-Editor-in-Chief

Paul Brindley

co-Editor-in-Chief

Reviewer's Responses to Questions

**Key Review Criteria Required for Acceptance?**

**Methods**

-Are the objectives of the study clearly articulated with a clear testable hypothesis stated?

-Is the study design appropriate to address the stated objectives?

-Is the population clearly described and appropriate for the hypothesis being tested?

-Is the sample size sufficient to ensure adequate power to address the hypothesis being tested?

-Were correct statistical analysis used to support conclusions?

-Are there concerns about ethical or regulatory requirements being met?

Reviewer #1: The objectives are clear, the study design is appropriate, the population is clearly defined, the sample size is adequate, statistical analyses were appropriate, no ethical concerns on my side.

Reviewer #2: (No Response)

**Results**

-Does the analysis presented match the analysis plan?

-Are the results clearly and completely presented?

-Are the figures (Tables, Images) of sufficient quality for clarity?

Reviewer #1: The analysis presented matches the analysis plan, the results are clear and completely presented, the figures are sufficiently clear.

Reviewer #2: (No Response)

**Conclusions**

-Are the conclusions supported by the data presented?

-Are the limitations of analysis clearly described?

-Do the authors discuss how these data can be helpful to advance our understanding of the topic under study?

-Is public health relevance addressed?

Reviewer #1: The conclusions are supported by the data presented, the limitations of the analysis are adequately described, the authors describe the relevance of the study.

Reviewer #2: (No Response)

**Editorial and Data Presentation Modifications?**

Reviewer #1: I believe the results are well presented and sufficient.

Reviewer #2: (No Response)

**Summary and General Comments**

Reviewer #1: All my previous comments were addressed and I have no further comments.

Reviewer #2: I think the edits/modifications are appropriate and help readers follow the flow of the models.

A small editorial suggestion is the provided definitions of preference, fidelity, and choice could be placed even earlier in the manuscript - line 52 introduces memory/fidelity with no clear meaning of these terms. I think lines 54-61 appropriately introduce the concepts of memory but still lack a definition to guide a reader. The PLOS NTD audience is familiar with choice and preference as concepts AND parameters but not necessarily with fidelity, so a clear conceptual explanation of fidelity needs to come before methods since it is central to the objective of the manuscript.

PLOS authors have the option to publish the peer review history of their article (what does this mean?). If published, this will include your full peer review and any attached files.

Reviewer #1: No

Reviewer #2: No